# Recent Advances on Heteroatom-Doped Porous Carbon—Based Electrocatalysts for Oxygen Reduction Reaction

Xue Li [1,†], Guolei Liu [1,†], Han Zheng [1], Kuizhao Sun [1], Linna Wan [1], Jing Cao [2], Saira Asif [2], Yue Cao [1,*], Weimeng Si [1], Fagang Wang [1,*] and Awais Bokhari [3,4,*]

1. Shandong University of Technology, Zibo 255000, China
2. Huantai Ecological Environment Management Service Center, Zibo 255000, China
3. Sustainable Process Integration Laboratory, SPIL, NETME Centre, Faculty of Mechanical Engineering, Brno University of Technology, VUT Brno, 616 00 Brno, Czech Republic
4. Department of Chemical Engineering, COMSATS University Islamabad (CUI), Lahore Campus, Lahore 54000, Pakistan
* Correspondence: cao-yue@foxmail.com (Y.C.); a_gang@sdut.edu.cn (F.W.); bokhari@fme.vutbr.cz (A.B.)
† These authors contributed equally to this work.

**Abstract:** Polymer electrolyte membrane fuel cells are considered one of the alternatives to fossil energy sources. The slow kinetics of the oxygen reduction reaction (ORR) at the cathode and the high price of Pt-based catalysts remain one of the key challenges for the commercial viability of proton exchange membrane fuel cells. However, their high cost and susceptibility to poisoning severely limit their use for large-scale commercial applications in fuel cells. Heteroatom-doped porous carbon has attracted extensive attention from scientists due to its advantages such as high specific surface area and the properties conferred by heteroatom doping. On the one hand, we discuss a variety of current methods for the preparation of heteroatom-doped porous carbons, including the template method and the activation method. On the other hand, we discuss the application of heteroatom-doped porous carbon in Pt catalysts, transition metal catalysts and metal-free catalysts. Finally, we also present the pre-existing and challenges of heteroatoms in ORR catalysis, which will drive the development of ORR catalysts.

**Keywords:** oxygen reduction; porous carbon; heteroatom doping; catalyst; specific surface area; platinum; transition metal

## 1. Instruction

With the growing concern about the energy crisis and environmental issues such as global warming, with traditional energy sources becoming depleted and new energy technologies insufficient to make up for major energy shortages, new technologies are becoming a hot research topic [1]. The exploration of clean energy and new energy storage technologies has become a new research hotspot. Clean and efficient energy conversion technologies, such as proton exchange membrane cells and metal cells, have received a lot of attention [2]. Oxygen reduction reaction (ORR) catalysts play an important role in reactions for proton exchange membrane cells and metal cells. The kinetics of oxygen reduction reactions involving multiple electron/proton transfer processes are very slow and require large amounts of electrocatalysts to accelerate the reactions [3]. Currently, platinum-based catalysts are still the most widely used commercial ORR catalysts with the best performance. Due to the scarcity of platinum, platinum-based catalysts account for about 60% of the cost in fuel cell stacks, which seriously hinders the commercialization of fuel cells. Therefore, reduced platinum usage and developing non-Pt electrocatalysts are two frontiers of efficient electrocatalysts for oxygen reduction [4,5].

Heteroatom-doped porous carbon is a traditional carbon material with microporous, mesoporous, and microporous multilevel pore structures [6]. Compared with conventional

materials, it has a rich pore structure multivacancy carbon has the advantages of high electrical conductivity, lower cost, and high solid density [7]. The pore structure greatly enhances the surface area of the material but also can enhance the adsorption capacity of carbon materials [6]. These nanomaterials with unique properties are of great interest to emerging nanotechnology researchers and play an important role in chemical capacitors, lithium-ion batteries, cationic batteries, hydrogen storage systems, photonic materials, fuel cells, and separation of toxic gases [6–8].

Although great progress has been made in the study of nanoporous carbon with various morphologies, pure carbon materials are poorly hydrophilic and do not have catalytic activity per se, thus limiting many potential applications, and doping of such materials with heteroatoms still has a wide scope for research [9]. When studying carbon materials, we should be aware that porous carbon materials are not only composed of carbon, but also doped with different heteroatoms, such as H, O, N and lesser amounts of S, B, P or halogen elements, during the synthesis process [10,11]. These heteroatoms are often doped as functional groups on the edges and surfaces of carbon materials [7]. Since the catalytic activity of doped carbon materials is influenced by the large number of functional groups containing heteroatoms, precise tuning of the heteroatom species and content is required to match the requirements of the application. In order to adapt porous carbon materials to different fields of application, simple methods to modulate the physicochemical properties of carbon materials need to be developed. Most of the current researchers base their efforts on two directions to modulate porous carbon materials: the construction of specific pore structures and the doping of different functional groups on the surface and inside the carbon material [12]. Heteroatom doping provides a useful tool to modulate and enhance the unique physicochemical properties of carbon, and doping not only changes the pore structure but also modulates the acidity and alkalinity of the material surface [13].

In this paper, we summarize the recent developments in electrocatalytic materials from porous carbon. In this review, we not only summarize various synthesis methods of heteroatom-doped porous carbon but also the mechanisms of different types of catalysts on ORR.

## 2. Synthesis Methods for NPCs

The methods for preparing doped multivacancy carbon are divided into two main categories: activation methods and template methods.

### 2.1. Preparation of Heteroatom-Doped Porous Carbon by Activation Method

Activation methods mainly refer to the preparation of materials by polymeric carbonization, physical activation, biomass carbonization and chemical activation. The size of the pore warp and pore structure of nanomaterials made by the activation method is usually more disordered [14]. It mainly includes the use of physical methods such as porogenic agents to create holes. Zhu et al. prepared 3D N-doped porous carbon materials (NDPC-X) by simultaneously carbonizing and activating $SiO_2$ layer-protected polypyrrole paper towels and pickling and etching them [15]. The activation method endows the material with abundant porosity structures, and the SBET of the material is as high as 1123.40 $m^2$/g. In an alkaline medium, the acquired NDPC-900 exhibited excellent oxygen reduction properties and action. In comparison with Pt/C ($-0.121$ V), the NDPC-900 catalyst shows a better reduction peak electric potential (0.068 V vs. Hg | $HgCl_2$) in oxygen reduction catalysis and exhibited better cycling stability and methanol resistance (as shown in Figure 1A). The chemical method includes pre-carbonation and other methods for chemical etching. A cobalt-carbon catalyst (Co-C) was prepared by an impregnation-precarbonation-activation strategy using natural fungus as a precursor by Lu et al. [16]. The fungus was first pre-carbonized at 350 °C and then activated by zinc chloride in an argon atmosphere at 1000 °C to produce a cobalt-carbon catalyst (Co-C). The acquired catalysts had several catalytic active sites, including N heteroatoms, essential defects, Co

nano particle and good dispersity CoNx structures, which enabled the Co-C catalysts to catalyze efficiently and stably in acidic, neutral and alkaline electrolytes oxygen reduction reactions with almost no degradation in performance after 5000 cycles in alkaline and neutral electrolyte stability tests (as shown in Figure 1B).

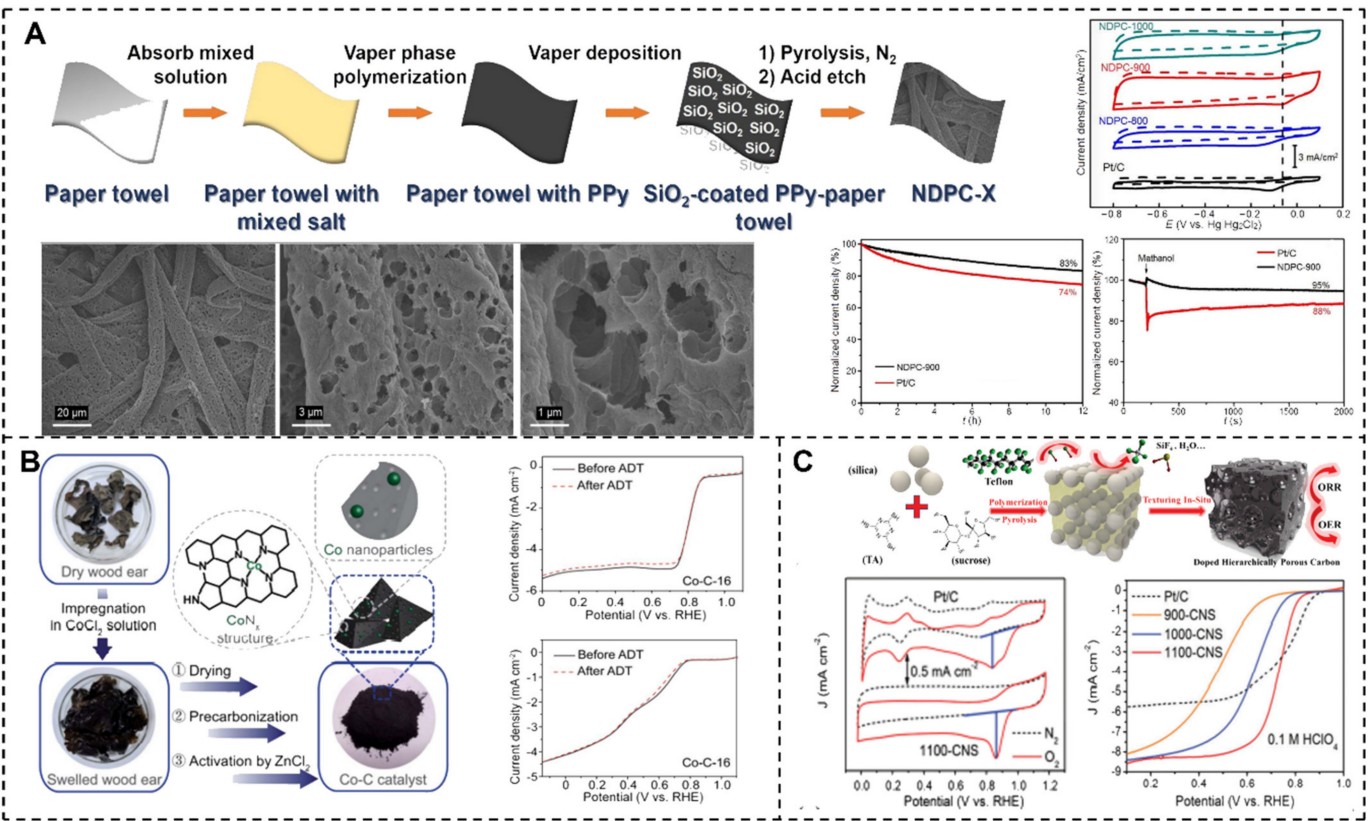

**Figure 1.** (**A**) Synthesis principle scheme, SEM images and electrochemical performance of NDPC-900 catalyst [15]; (**B**) Schematic diagram of Co-C catalyst preparation using wood ear as a precursor and durability test of catalyst [16]; (**C**) Diagram of the one-pot preparation process of doped porous carbon materials and CV, LSV curves of the Pt/C and the metal free samples catalyst [17].

## 2.2. Preparation of Heteroatom-Doped Porous Carbon by Template Method

The stencil approach can be split into the soft stencil approach, the hard stencil approach and the double stencil approach. The template method can produce the same material with the same hole structure in batches and on a large scale [18]. One-dimensional porous Fe/N-doped carbon nanorods (Fe/N-CNR) with graded micro/mesoporous structures were prepared by Chen et al. The $Fe_2O_3$ nanorods were not only partially dissolved to generate $Fe^{3+}$ for initiating the polymerization reaction, but also formed 1D structures as templates during the polymerization process [16]. In addition, the pyrrole-coated $Fe_2O_3$ nanorod structure prevents the collapse of the porous structure and protects Fe from aggregation, thereby producing an atomic $Fe-N_4$ structure during the carbonization process. The obtained Fe/N-CNR exhibited excellent Oxygen Reduction Reaction activity ($E_{1/2} = 0.90$ V) and satisfactory longtime durability, exceeding that of Pt/C. As shown in Figure 1C, a multistage porous carbon skeleton catalyst rich in N, S, as well as O elements, was prepared by in situ chemistry by Zhi et al. at the City University of Hong Kong [17]. The researchers obtained porous carbon catalysts with macroporous-mesoporous-microporous structure, large specific surface area and rich in heteroatom doping by a simple one-step calcination method using $SiO_2$ of appropriate size as template, inexpensive sucrose as carbon source, common sulfide cross-linker trithiocyanate as N and S source; The method wards the use of highly noxious HF reagents or strong alkaline solutions as template etching agents, and

eliminates the need for post-purification or further activation processes, thus avoiding the loss of active sites and simplifying the synthesis of the catalyst. The catalyst exhibits excellent Oxygen Reduction Reaction performance in both acidic ($HClO_4$, $H_2SO_4$) basic and basic (KOH) solutions, exceeding most of the reported metal-free and metallic catalysts.

## 3. Application of Heteroatom-Doped Porous Carbon in Pt Catalysts

Heteroatom-doped carbon-loaded Pt-based catalysts have successfully solved the problem of low Pt utilization in terms of electronic structure. The heteroatom-doped carbon-loaded Pt catalysts have well-defined active sites, low coordination environment, ultra-high atomic utilization, and strong interactions between the heteroatoms and individual metal atoms to both modulate the electronic structure and increase the stability of the catalysts, thus achieving high catalytic activity and selectivity. An ultra-low Pt loading (1.1 wt.%) carbon-loaded Pt single-atom dispersion catalyst with high efficiency was prepared as a carrier [19]. The results show that the introduction of defects achieves an efficient and highly stable dispersion of Pt, which greatly improves the utilization of Pt in fuel cells. In addition, a significant raising in the surface area of the carrier material exposes the largest number of active sites to contact the reactants, which is considered an effective way to increase the intrinsic activity of these loaded unit site catalysts. Madhumita et al. reported a novel catalyst of sulfur co-doped porous carbon-loaded Pt nanoparticles (Pt/DPC) synthesized by solvation using 1-ethyl-3-methylimidazolium dicyandiamide ionic liquid as a precursor and eutectic salt as a pore expander [14]. It was found that because of the doping of heteroatoms the nucleation and growth kinetics of Pt nanoparticles in the process of catalyst deposition were altered, which led to a reduction in catalyst particle size and an enhancement in Pt particle dispersion. The Pt/DPC was tested to exhibit good oxygen reduction activity in acidic solutions and showed excellent stability.

As shown in Figure 2A, Wang et al. of Hong Kong Polytechnic University, prepared hollow multihole nitrogen-doped carbon spheres (rich mesopores) encapsulated with Co nanoparticles (Co@HNCS) precursors for the first time by cobalt pre-embedding and pursuant impregnation-reduction means, and obtained O-PtCo3@HNCS by subsequent impregnation and high-temperature treatment [20]. The Co pre-embedding step formed rich mesoporous and the impregnation-reduction strategy led to the ordering of Pt/Co and carbon encapsulation. Thanks to the ameliorative mass transfer process, reinforced metal interactions and physical confinement effects, O-PtCo3@HNCS reveals good ORR performance and durability in acidic solutions. The nitrogen-doped carbon shell with a hollow structure acts as a protective shell and the ordered $PtCo_3$ nanoparticles are tightly immobilized in the carbon matrix. It not only inhibits sintering in the high-temperature ordered process, but also suppresses agglomeration, clustering and metal leaching in the electrochemical potential cycle.

In order to synthesize high-performance biomass carbon catalyst carriers, members of this group prepared a biomass carbon material loaded with Au@Pd@Pt catalyst by a green and simple method in response to the problems of existing Pt-based catalysts that tend to agglomerate and fall off in the electrolyte (as shown in Figure 2B) [21]. That is, using tofu gel as the raw material, after high-temperature pyrolysis of proteins and other biomass in tofu, the tofu biomass carbon (NSC) was produced and successfully loaded with Au@Pd@Pt catalyst. N and S heteroatoms in NSC can effectively immobilize the catalyst, and Au@Pd@Pt-NSC-900 after 20,000 cycles, the material mass activity loss was 4.9% and E1/ The doping of N and S in NSC effectively prevents the agglomeration, exfoliation, and ripening of Pt, which is the major reason to ensure ORR stability of Au@Pd@Pt-NSC-900. In addition, the Au@Pd@Pt dendritic structure also improves the utilization of Pt. Through the good synergy between NSC and Au@Pd@Pt, a catalyst with low Pt and high ORR performance can be obtained.

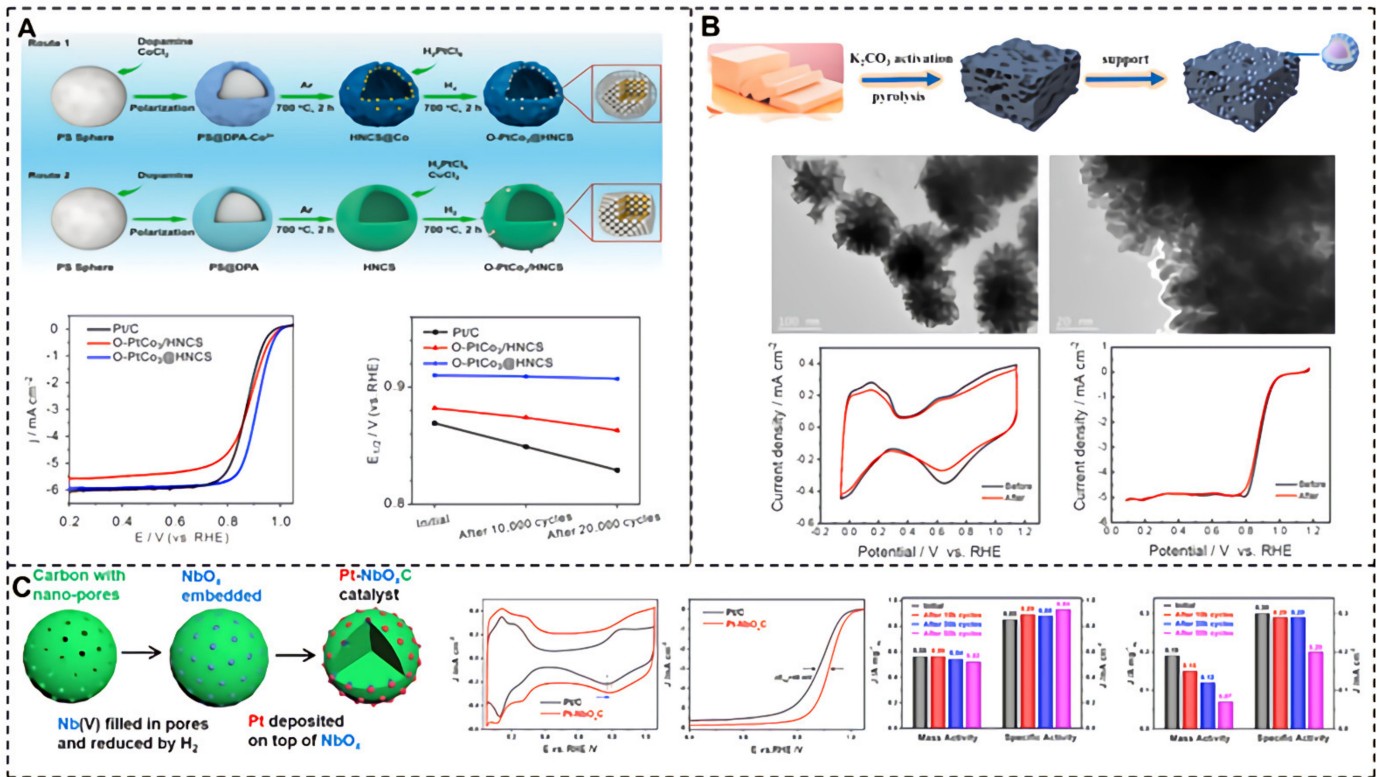

**Figure 2.** (**A**) Elementary diagram of the synthesis of O-PtCo3@HNCS and O-PtCo3/HNCS and electrocatalytic properties of Pt/C, O-PtCo3/HNCS, and O-PtCo3@HNCS [20]. (**B**) Elementary diagram of the Au@Pd@Pt-NSC catalysts and the TEM images, CV curves, polarization curves of the Au@Pd@Pt-NSC-900 material [21]. (**C**) Schematic illustration of Pt/metal oxide/carbon composite catalyst (Pt-NbOxC) and cyclic voltammetry curves, ORR polarization curves [22].

Wei et al. successfully loaded sub-3 nm high-entropy Pt alloy nanocrystals (HENs) onto the carbon skeleton using a one-step anchoring-carbonization strategy, which resulted in effective planar bonding between HENs and the porous carbon skeleton by first anchoring the nanocrystals to the partially carbonized porous carbon skeleton in the medium-temperature region [23]. In situ carbonization around the HENs in the high-temperature region not only suppressed the size growth of the HENs to a large extent, but also allowed the HENs to form more effective planar contacts with the porous carbon and obtain strong interfacial interactions. This method not only increases the anchoring strength between HENs and carbon skeletons, but also effectively suppresses the movement and size increase of HENs during the carbonization process. The novel catalytic material porous carbon-loaded hexameric high-entropy Pt alloy nanocrystals (6-HENs-PC) exhibited excellent ORR catalytic activity, with 3.8 times the mass activity of commercial Pt/C at 0.9 VRHE, while the half-wave potential and high-entropy structure remained almost unchanged after 5000 cycles.

Chen et al. proposed new Pt/metal oxide/carbon catalysts with a hierarchical structure (Figure 2C) [22]. The authors first used porous carbon to anchor small particle size, low valence NbOx and then selectively deposited Pt on the NbOx surface. This strategy effectively reduces the direct contact between the metallic platinum and the carbon carrier and slows down the catalytic corrosion of the platinum on the carbon material. In the meantime, the strong interaction between the metal oxide and Pt layer enhances the catalytic ability of Pt atoms and improves the problems of agglomeration and shedding of Pt particles. After stability testing, the mass-to-performance of Pt-NbOxC reduced by only 7% after 50,000 turns of accelerated aging experiments. Besides, the small size and low valence of NbOx nanoparticles and their close contact with the carbon carrier effectively heighten

the overall conductivity of the catalyst and enhance the oxygen reduction catalytic ability of the material.

## 4. Application of Porous Carbon Materials Doped with Foreign Atoms in Metal-Free Catalysts

Huge efforts and financial resources have long been invested in the research of non-noble metal electrocatalysts. Heteroatom-doped carbon has relatively high activity and four-electron selectivity in acid, and thus is considered the most promising non-noble metal electrocatalyst to displace Pt-based materials. The research on heteroatom-doped carbon ORR catalysts traces back to 1999, when researchers synthesized the first class of nitrogen-doped CNTs by heating iron phthalocyanine (FePc) to decompose [24]. Nitrogen residues in CNTs were long considered undesirable "impurities" until Dai et al. developed the first nitrogen-introduced carbon nanotubes (N-CNTs) as beneficial ORR electrocatalysts for alkaline oxygen reduction reactions (ORR) in 2009 [25]. Experimental data and computational analysis together demonstrate that the catalytic activity of heteroatom-doped carbon is caused by heteroatom-induced charge transfer: some electrons of carbon atoms are transferred to their neighboring, more electronegative heteroatoms, leading to a change in the chemisorption pattern of $O_2$ molecules and shorten of the "O-O" bond, resulting in catalytic ORR performance. This pioneering work explained for the first time the catalytic mechanism of heteroatoms and pushed the research boom worldwide.

Due to the chemical instability of transition metal-based and N-doped catalysts in acidic solutions, such catalysts are susceptible to degradation or loss of catalytic performance due to the deactivation of the catalyst's active sites when tested in acidic media for long periods of time [26,27]. In addition, the study also found that the protonation of the doped nitrogen atoms hinders charge delocalization due to the change in the active site of the nitrogen-doped carbon material in both media and its interaction with oxygen, thus reducing its electrocatalytic activity in acidic media [26]. Since different heteroatoms have different radii and abilities to attract electrons, their degree of modulation of the electronic structure of the matrix carbon may also vary. Qiao et al. explored the catalytic activity of graphene doped with different heteroatoms and noticed that the ORR catalytic performance of these heteroatom-introduced carbons is highly dependent on the properties of the dopant atoms themselves, which means that different heteroatom dopants exhibit completely different electrochemical Theoretical calculations predict the optimal heteroatom [28]. Theoretical calculations predict that the exchange current density of the optimized heteroatom-doped graphene is much better than that of Pt/C catalysts, implying that heteroatom-doped carbon is promising as a cost-effective catalyst to replace platinum-based noble metals.

Defects in the graphite structure have also been found to alter the altered electron distribution of carbon atoms and play an integral role in the modulation of heteroatom-doped carbon materials. Edge defects in carbon nanomaterials, such as dangling bonds and hydrogen-saturated bonds, lead to charge transfer between carbon atoms at different locations (at the edges or in-plane) as well, providing active sites for electrochemical catalytic reactions. Wang's treatment of polypyrrole nanotubes by potassium permanganate etching introduced a large number of pore structures and produced many defects in the carbon materials, providing favorable conditions for pyridine-N formation [29]. Meanwhile, the oxygen-containing functional groups formed in the etching process continue to increase the number of defects in pyrolysis, which increases the number of edge N atoms and improves pyridine N concentration, which the content in the carbon material is increased from 32.8% to 45.2%. The improved material has better half-wave potential than Pt/C under alkaline conditions and possesses an energy density of 835 Wh kg$^{-1}$ with a peak power of 122 mW cm$^{-2}$ when the catalyst is used in a zinc-air cell (as shown in Figure 3A). Jiang et al. used N-introduced carbon nanotubes coated with cobalt particles as a model catalyst [30]. The characterized results show that the nitrogen-doped carbon layer has many structural defects that reduce the electron transport resistance. The half-cell test results

revealed that the synthesized catalysts have better ORR performance than Pt/C and exhibit excellent catalytic stability. Theoretical calculations show that the graphite-N-doped carbon nanotube-coated cobalt carries more negative charges than the pyridine-N-doped carbon nanotube-coated cobalt, indicating that more electrons occupy the anti-bonding orbitals of oxygen, which can effectively activate the $O_2$ molecules. Assembled into a zinc-air cell, the catalyst exhibited superior activity to the Pt/C having an output energy density of 837 Wh kg$^{-1}$ (Figure 3C).

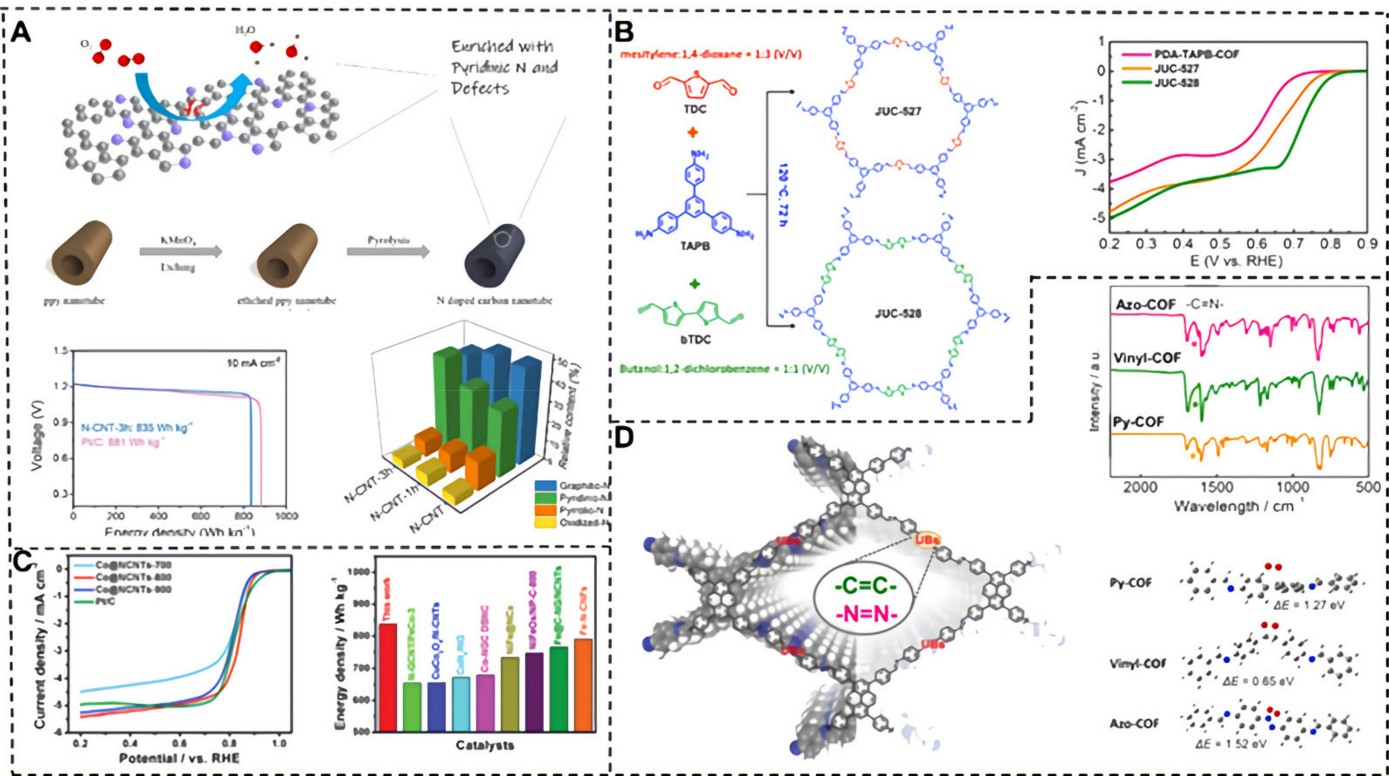

**Figure 3.** (**A**) Schematic diagram of the prepared N-doped carbon nanotubes and energy density of Zn-air battery [29]. (**B**) Structures and synthetize of JUC-527 and JUC-528,and the LSV curves of JUC-527, and JUC-528 [31]. (**C**) LSV curves for the ORR, and the energy density of Co@NCNTs-800 and other catalysts [30]. (**D**) The unsaturated bond decorated organic frameworks, and FT-IR spectra, and the results of oxygen binding energy calculation of Azo-COF Vinyl-COF and Py-COF models [32].

In the last decade, metal-free carbon materials (MFCM) accompanied by defects or the introduction of foreign atoms have been considered replaceable catalysts for oxygen reduction reactions (ORR). Yao et al. exploited and prepared stable MFTS COFs by linear thiophene-S building blocks as active catalysts for ORR acceleration (as shown in Figure 3B) [31]. It was demonstrated that COFs with more thiophene-S structures provided better ORR performance, and DFT calculations reasonably indicate that thiophene-S structures are active sites. The $O_2$ and intermediate travel distances in the porous structure are shortened. the linear heterocyclic building blocks of MFTS COF are able to finely tune the ORR performance.

Yang et al. prepared a variety of metal-free COFs modified with unsaturated bonds using pure carbon monomers and precisely revealed the fixed molecular backbone, modulable electronic structure and ORR dynamics [32]. Experimental and theoretical calculations show that the modification of unsaturated bonds on COF increases the adsorption of oxygen intermediates and reduces the LUMO energy, and the inherent UBS can attract local charge redistribution, which gives the molecular backbone a high surface map distribution and an effective attraction for oxygen intermediates and optimizes catalytic reaction process (Figure 3D). In addition, the advantageous thermodynamics of the para-C=N group

affected by unsaturated N=N conferred the efficient ORR catalytic activity of Azo-COF. The presence of the active center was further proved through in-situ Raman spectroscopy.

### 5. Application of Heteroatom-Doped Porous Carbon Hetero-Carbon Materials in Transition Metal Catalysts

Besides Pt-based catalysts, transition metal catalysts are another option, but their poor stability and low electrical conductivity cannot meet the practical requirements. The introduced heteroatomic energy effectively modulates the carbon materials' electronic structure and physicochemical properties, thus affecting the adsorption detachment of oxygen-containing substances, which is an effective strategy to improve the catalytic performance of transition metals [33,34]. Heteroatom-doped carbon-loaded transition metals excess metal catalytic materials have been widely used for ORR electrocatalysts due to their similar activity, and better stability compared to noble metal-based catalysts (Figure 4B) [33,35,36]. Compared to Pt/C, the optimized Fe-ISA/SNC shows a half-wave potential of 0.896 V. XAFS analysis and DFT calculations shows that the relatively low electronegativity of S enriches the charge on the N atom, facilitating the decisive step OH* reduction release and thus accelerating the overall ORR process. A N and P double-liganded single-atom iron composite carbon nanosheet (Fe-N/P-C) was investigated by Chen et al. [37]. The chemical coordination structure of the Fe-N3P center was determined by XANES and EXAFS, and this catalyst with abundant Fe-N3P active sites exhibited excellent catalytic performance durability and methanol tolerance for ORR, and outperformed Pt/C because of the compositional and structural advantages and the optimal electronic structure of the active center at the atomic level. Calculations showed that the N P double-coordinated iron sites facilitate the adsorption/desorption process of oxygen intermediates, which can accelerate the kinetics of ORR and achieve the high catalytic activity.

Zhao et al. designed heterogeneous single-atom ORR electrocatalysts in which nitrogen-ligated Fe- and Ni-mono atoms were co-doped on an ordered graded porous carbon carrier consisting of highly ordered macropores with interconnected micro/mesopores [4]. XANER and EXAFS confirmed that Fe- and Ni-single atoms were anchored to the carbon carrier through Fe-N4 and Ni-N4 coordination bonds. The prepared catalysts exhibited excellent ORR activity, which was superior to electrocatalysts containing only Fe- or Ni- single atoms and the benchmark Pt/C catalysts. Calculations showed that the catalytic activity was due to the synergistic effect of coexisting Fe-N4 and Ni-N4 sites, while the ordered porous carbon carriers promoted the mass transfer capability of the materials (Figure 4A).

Ji et al. successfully constructed nitrogen-doped porous carbon-loaded Fe monatomic catalysts by Schiff base polymerization of aminoporphyrin, aminoporphyrin, and terephthalaldehyde followed by high-temperature carbonization under a precursor dilution strategy [38]. The Fe monoatoms were determined to be dispersed in the carrier at the atomic level in the FeN4O structure by combining with spherical differential electron microscopy and synchrotron radiation characterization. The Fe monatomic catalysts in alkaline electrocatalytic oxygen reduction exhibited superior performance in catalytic activity, cycling stability, and tolerance to methanol compared to Pt/C.

Ge et al. prepared Fe-doped Fe-ZIF-L by a two-solvent method using Zn-ZIF-L with a bladed morphology as a precursor, and then obtained nitrogen-doped porous carbon materials (Fe-L-CNT-900) coated with carbon nanotubes by two-step high-temperature pyrolysis [39]. The two-dimensional porous nitrogen-doped material was used as a carrier, which enabled Fe-L-CNT-900 to have the maximum specific surface area, abundant micro-mesoporous structure, high graphitization degree, moderate heteroatom doping, and exhibited the best electrocatalytic oxygen reduction performance with the initial potential and half-wave potential up to 1.040 V and 0.88 V, which were better than the Pt/C.

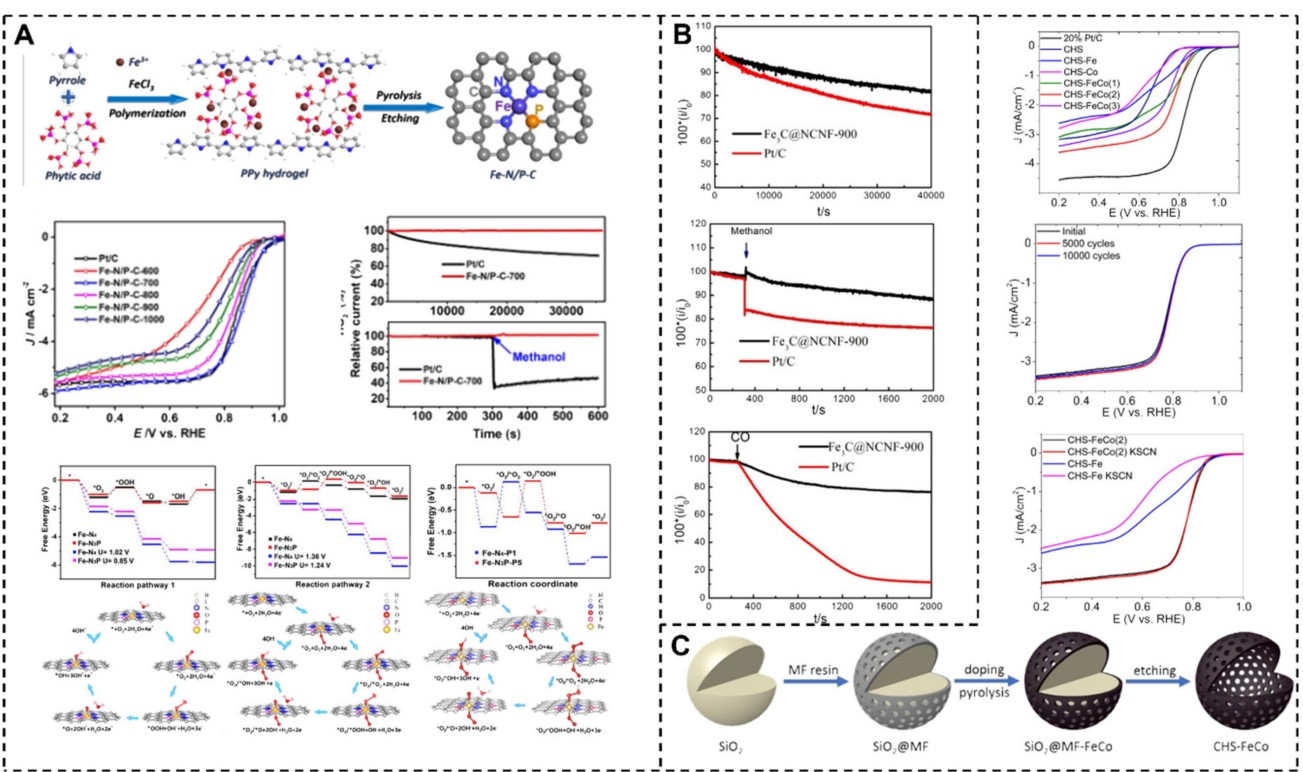

**Figure 4.** (**A**) Schematic illustration of the synthesis process of the Fe-N/P-C catalyst, ORR polarization curves of Fe-N/P-C, tolerance to methanol and current−time chronoamperometric responses and the Gibbs free energy diagrams of ORR on Fe-N4 and Fe-N3P active sites [34]. (**B**) Durability evaluation from the current-time chronoamperometric responses, and i-t curves for CO-poison and the methanol-crossover effect [30]. (**C**) Elementary diagram of the preparation of CHS-FeCo, and polarization curves of different CHS instances, and stability test of CHS-FeCo, and ORR polarization curves with KSCN added [37].

Chen et al. at the University of California, Santa Cruz, USA, developed an iron-cobalt bimetallic center with nitrogen-co-doped carbon nanocages and found that the introduction of the nanocages can effectively protect the catalytically active center with significant resistance to KSCN toxicity and excellent catalytic activity for ORR in alkaline environments [40]. The team used $SiO_2$ nanospheres as a template, polymerized melamine-formaldehyde resin as a shell layer, and added different kinds of iron and cobalt precursors for pyrolysis and etching reactions to prepare the carbon nanocages (Figure 4C). In the electrochemical characterization, the catalytic activity of the bimetallic-doped carbon cages was significantly higher than that of the sample containing only a single metal; the best sample, CHS-FeCo, exhibited an initial potential of up to +0.93 V and a half-wave potential of +0.79 V, as well as high stability and resistance to KSCN toxicity.

Li et al. utilized dopamine as a carbon and nitrogen source precursor, in which Fe ions were fixed in the polymerization process, and additionally introduced the block copolymer F127 to achieve the coexistence of microporous and mesoporous (Figure 5A) [41]. The resulting catalyst has a unique mulberry-like porous structure with excellent continuous electron transport and ion diffusion capability and a specific surface area of 913.2 $m^2$/g. The catalytic has a half-wave potential of 0.85 V and an ultimate current density of 5.35 mA $cm^{-2}$, which even surpasses the commercial Pt/C catalysts, showing quite excellent catalytic activity. The data showed that the catalyst reduced oxygen directly to water via a 4-electron pathway with high selectivity. In addition, the Fe-, N-, and S-co-doped porous carbon structures exhibited extremely high stability and methanol resistance, far exceeding commercial Pt/C catalysts.

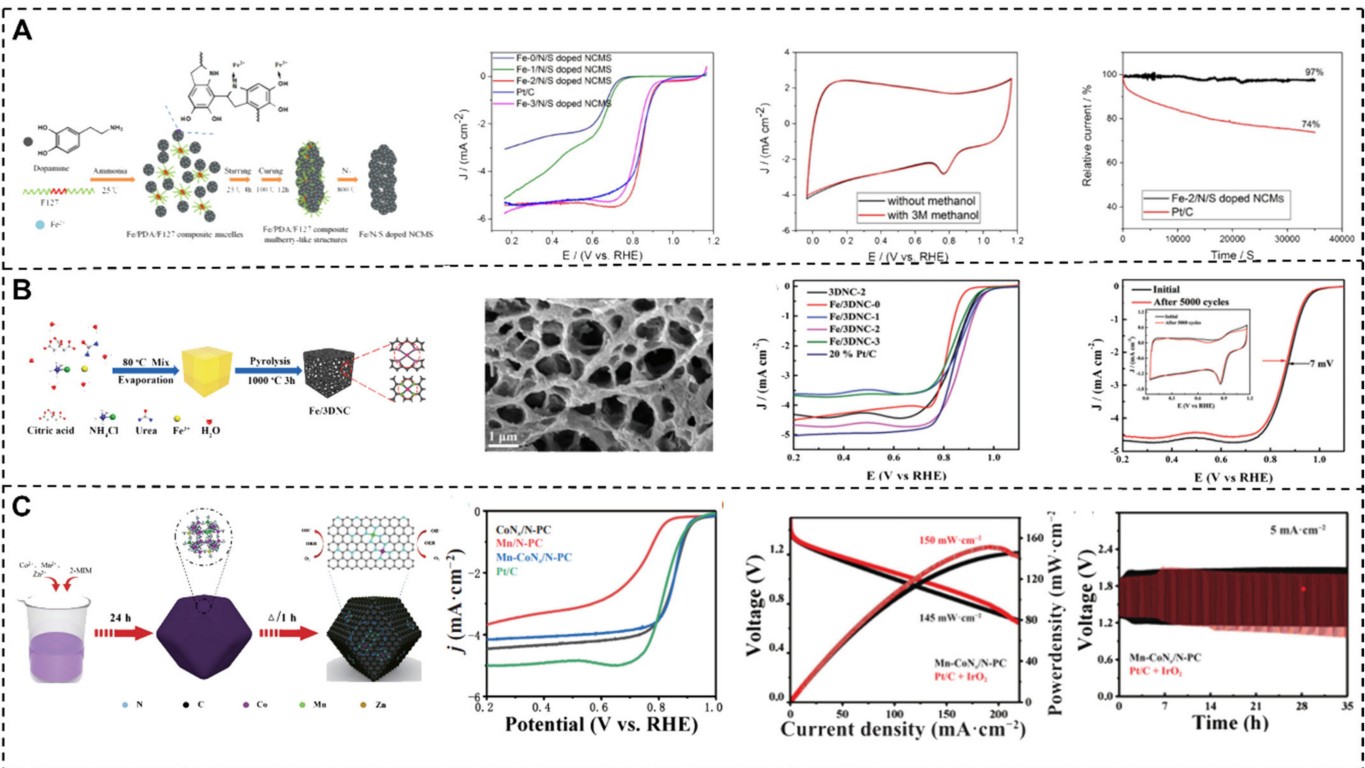

**Figure 5.** (**A**) Schematic illustration of the preparation of the Fe/N/S-doped NCMs, and LSV curves, and methanol tolerance tests, and chronoamperometric response [41]. (**B**) Schematic illustration of the preparation of the Fe/3DNC, and SEM images, and LSV curves, and comparison of ORR polarization curves before and after 5000 cycles [42]. (**C**) The preparation process of Mn-CoNx/N-PC, ORR polarization curves, the power density and charge-discharge testing of Mn-CoNx /N-PC [43].

In porous carbon, the introduction of heteroatoms is important for enhancing oxygen reduction properties. A shell carbon nanosphere was prepared by Lu et al. at Huazhong University of Science and Technology using fullerene C60 as a carbon precursor and reacted with ethylenediamine, which could be transformed into defect-rich N-doped porous shell sphere structures (N-PHCNSs-800) after high temperature (800 °C) pyrolysis, and further in the presence of sublimated sulfur, this shell carbon nanosphere could be transformed into N/S co-doped shell sphere structures (N,S-PHCNSs) [44]. The electrochemical test date showed that different amounts of sulfur introduction had a significant effect on the catalytic activity of N, S-PHCNSs. With the increase of sulfur introduction, the N, S-PHCNSs catalyst's activity increased and then decreased, among which N,S-PHCNSs-75 had the best catalytic activity for ORR and comparable to Pt/C catalysts. Density flooding calculations show that the graphite N and thiophene S co-doped five-membered ring defect has a unique electronic structure with a four-electron reaction decisive step different from several other carbon structures, and overcomes the smallest energy barrier and exhibits high catalytic activity, which is consistent with the experimental results.

Li et al. prepared a nano iron-modified nitrogen-doped three-dimensional porous carbon framework catalyst by using a rapid gas foaming method. The micromorphology and catalytic efficiency were controlled by adjusting the amount of urea [42]. The optimized Fe/3DNC-2 catalyst has a hierarchical porous structure (Figure 5B) with a specific surface area of 438 $m^2 \cdot g^{-1}$, and uniformly distributed active centers (pyridine N, graphite N and FeNX). Due to the three-dimensional structure and the doping of pyridine N and graphite N, the Fe/3DNC-2 has excellent activity and durability for ORR, such as half-wave potential (0.874 V) and initial potential (0.995 V), which are higher than commercially Pt/C and the half-wave potential shifted by about 7 mV after 5000 cycles.

Peng et al. proposed a MnO-doped CoNx/N-PC composite (Mn-CoNx/N-PC) prepared from carbonized metal-organic backbone (MOF) derivatives as a bifunctional (ORR and OER) electrocatalyst to improve catalytic performance [45]. The doping of N can significantly enhance the catalytic activity of Co-N-C for the oxygen reduction reaction center. At the same time, the doping of Mn effectively regulates the electronic structure of the Co element, increasing the content of Co0, thus providing an effective OER active center. The experimental results show that the Mn-CoNx/N-PC catalyst has superb bifunctional electrocatalytic performance with a half-wave potential of 0.85 V, which is superior to 0.82 V of Pt/C (as shown in Figure 5C). Further, the Mn-CoNx/NPC electrocatalyst was used for zinc-air batteries with a power density of 145 mW cm$^{-2}$ and good cycling performance.

Compared with other carbon carriers, MOF-derived nanostructures have diverse structures, tunable conformations, uniformly distributed pores and multidimensional morphologies, which can be fully controlled by adjusting the linker types and metal centers of MOF precursors. Wang et al. prepared Fe (II)-ZIF-8 under mild synthetic conditions, followed by a one-step thermal treatment to obtain Fe(II)-N-C structured catalysts with high dispersion of active sites [46]. The dispersive materials with high specific surface area and stable 3D structure were prepared by using the properties of MOF. In addition, the use of MOF as a carrier also improved the dispersion of FeN$_x$ and enhanced the ORR catalytic activity. The Fe-N-C structured catalyst completed the ORR reaction through a four-electron process with an onset potential and half-wave potential of 0.95 V and 0.82 V vs. RHE, respectively, under acidic conditions, and exhibited excellent stability and methanol resistance.

Different types of nitrogen also affect the performance of oxygen reduction [43]. The high content of pyridine and graphitic nitrogen can improve the ORR catalytic ability of NSC, especially graphitic N can significantly increase the ultimate current density and accelerate the charge transfer [47]. Moreover, it is known after DFT calculation that graphitic nitrogen can effectively reduce the energy barrier from O$_2$ to *OOH in the ORR reaction [48].

Qiao et al. designed and prepared a nitrogen-rich several-layer graphene structure using a simple g-C$_3$N$_4$ template method and achieved selective tuning of pyrrole nitrogen doping by changing the ratio of raw materials [49]. It was found that the content of pyrrole nitrogen in the material was proportional to the selectivity of its hydrogen peroxide synthesis. The quasi-in situ XANES test revealed that pyrrole nitrogen plays an important role in the adsorption of two-electron ORR intermediates. The high pyrrole nitrogen content and the unique fluffy and porous structure of the material resulted in excellent hydrogen peroxide electrosynthesis performance. Finally, by combining biomass conversion (furfural oxidation) reactions, the assembled practical devices can achieve the synthesis of high-value chemicals simultaneously at the cathode and anode at a small voltage.

In terms of mechanism, although still imperfect, it is widely believed that doping N, S, B, and P into the carbon skeleton significantly induces changes in the electron cloud of C atoms attached to the heteroatoms, forming tiny activation regions, and such activation regions provide carbon materials with many unprecedented oxygen reduction catalytic properties [50]. Many researchers have demonstrated theoretically and experimentally that pyridine-N has higher performance compared to other configurations (pyrrole-N, graphite-N and oxide-N). Binary (N/S, N/P, N/B, etc.) or ternary heteroatom (N/S/P, N/B/P, etc.) doping is intended to improve the activity by modulating the local electronic environment and increasing the sensitivity of the active center. However, the arbitrary distribution of polyheteroatoms on carbon substrates limits the synthesis and exploration of specific polyheteroatom configurations with optimal activity [51]. Fan et al. developed a one-pot "two-catalyst" strategy to successfully dope pyridine N-B pairs into carbon carriers while forming an amorphous boron (GNs)/amorphous carbon hierarchy in situ [52]. During the synthesis, high concentrations of H$_3$BO$_3$ were coupled with Fe(III) salts to form 2D graphene-like layers (denoted as FeBO layers). During high-temperature annealing, B atoms can absorb electrons from C atoms, leading to C-C bond breakage, which in turn leads to the rearrangement of the carbon skeleton. Fe(III) salts are another catalyst for

the formation of carbon microcrystals by the mechanism of C infiltration into metallic Fe followed by C precipitation. The formed carbon microcrystals are interconnected on the FeBO layer (as a hard template), leading to the growth of GNs.DFT simulations reveal that the pyridine N-B pair has the highest catalytic activity among all potential configurations due to the highest charge density at the C active site adjacent to B, which enhances the strength of the interaction with the p-band center intermediate.

## 6. Summary and Outlook

As ORR catalysts have been intensively investigated and potential low-platinum and platinum-based catalyst alternatives have been explored, a series of heteroatom-doped porous carbons have been investigated for the preparation of highly stable, highly active ORR catalysts. These catalysts not only reduce the cost but also improve fuel cell efficiency [52]. In-depth studies of heteroatom-doped porous carbons and their intrinsic mechanisms will definitely contribute to further improvement of ORR catalytic kinetics and reduction of Pt catalyst usage. This paper focuses on the synthesis of different heteroatom-doped porous carbons and the application of heteroatom-doped porous carbons in Pt-based catalysts, over-metallic catalysts and metal-free catalysts. These catalysts demonstrate excellent catalytic activity in ORR tests and open up new avenues for the construction of high-performance fuel cells.

We likewise present the application of cutting-edge heteroatom-doped porous carbon in Pt-based catalysts, transition metal catalysts and metal-free catalysts. According to recent progress, heteroatom-doped porous carbon catalysts exhibit excellent ORR catalytic activity. However, there are still many problems to be solved in the process of further improving the catalyst performance.

1. The stability of heteroatom-doped carbon materials in acidic media is an issue due to the protonation or oxidation of heteroatoms in the acidic environment.

2. The interaction relationship between heteroatoms in heteroatom-doped porous carbon and the catalytic mechanism of heteroatoms on catalytically active centers is still in doubt. The synthesis of decorated porous carbon nanostructures usually requires high-temperature carbonization in a specific gaseous environment, which adds difficulty to the mechanistic interpretation.

3. Controlling the pore structure and size of biomass-derived NCs is challenging. Although the conventional template method can prepare materials with uniform pore structure, it cannot be produced on a large scale.

In conclusion, further research is needed in the quest for high-performance and highly stable heteroatom-doped carbon catalysts, and heteroatom-doped porous carbon materials are highly anticipated in ORR catalysts. It is expected that the synthetic methods explored in this review and their applications in various catalyst fields will contribute to the development of heteroatom-doped porous carbon and trigger more innovations to stimulate ORR catalyst research and industry.

**Author Contributions:** Conceptualization X.L.; Methodology X.L., G.L. and H.Z.; Investigation X.L. and J.C.; Visualization X.L., G.L. and L.W.; Data Curation X.L., G.L., H.Z. and K.S.; Writing—Original Draft X.L.; Supervision S.A., Y.C., W.S., F.W. and A.B.; Writing—review & editing. F.W., Y.C. and A.B. All authors have read and agreed to the published version of the manuscript.

**Funding:** This work was supported by the National Natural Science Foundation of China (51502161, 51572127, and 21576138), the Natural Science Foundation of Shandong Province (ZR2014EMQ008).

**Institutional Review Board Statement:** Not available.

**Informed Consent Statement:** Not available.

**Data Availability Statement:** Not available.

**Conflicts of Interest:** The authors declare no conflict of interest.

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
