# Peer review of "Recent Advances on Heteroatom-Doped Porous Carbon—Based Electrocatalysts for Oxygen Reduction Reaction"

_energies, doi:10.3390/en16010128_

Round 1
Reviewer 1 Report
Comments:
In this current review article by Liu et.al. tried to put light on recent advances on Heteroatom-doped porous carbon -based Electrocatalysts for Oxygen Reduction (ORR). The article is encouraging, however, it is poorly written. The texts need a lot of correction and rearrangements. The paper also lacks recent works on hetero atom doping for ORR.
There are few issues which I think the authors should address before being considered for publication.
As the paper focuses on the synthesis of different heteroatom-doped porous carbons and the application of heteroatom- doped porous carbons in Pt-based catalysts, over-metallic catalysts and metal-free catalysts. Therefore, in synthesis and application part authors should include the following points.
1. Authors should describe recent works using MOF as template which stands to be one of the major advances for high surface area electrocatalysts and cite relevant literature.
2. Self-templating synthesis of porous carbons should also be discussed and cited properly.
3. There is a lot typos and grammatical mistakes in the manuscript, which needs to be rechecked and corrected. E.g. 1. Instruction rather than Introduction, Density flooding calculation (page 12 line 466) to be mentioned a few.
4. References are not formatted according to journal standard. It even shows invalid citation??? Citation 14. So please check and correct them.
Authors should increase the legibility of the figure for a better view and understanding.
Author Response
Many thanks to the reviewers for their comments, which we have carefully responded to, details of which are attached.

Reviewer 2 Report
The manuscript reported the heteroatom-doped porous carbon with a high specific surface area, including the template method and the activation method. The application of heteroatom-doped porous carbon in Pt catalysts, transition metal catalysts and metal-free catalysts in PEMFCs are summarized and reviewed. The pre-existing challenges of heteroatoms in ORR catalysis, which will drive the development of ORR catalysts are also discussed.
I consider the content of this manuscript will definitely meet the reading interests of the readers of the Energies journal. However, there are certain English spelling and grammar issues, and also the discussion and explanation should be further improved.
Therefore, I suggest giving a minor revision and the authors need to clarify some issues or supply some more experimental data to enrich the content. This could be comprehensive and meaningful work after revision.
Detailed comments can be found in the PDF file.

Author Response

(The authors gave the same response as above.)

Reviewer 3 Report
The manuscript entitled “Recent advances on Heteroatom-doped porous carbon-based Electrocatalysts for Oxygen Reduction Reaction” consists of a comprehensive review of carbon-based electrocatalysts towards the ORR. This is a well-written manuscript (though some minor errors can be found in writing) with a brief overview of the recent experiments in carbon materials. Nevertheless, the title of the work mentions “heteroatom-doped”, while in the manuscript, there is no special attention to this type of electrocatalysts but in metal-containing electrodes. Before publishing, this work needs to address the following important points.
1. Re-check the writing of the manuscript. Although is well written, there is still important errors in the work. Some examples:
a. Use subscript for O2, P332
b. Use capital letters after points, P333
2. After reading the title and abstract, I was expecting some more detailed information about heteroatom-doped metal-free carbon materials. However, the manuscript focuses more on metal-containing electrocatalysts. It is necessary to include more information about the different heteroatoms that can tailor the surface chemistry of the carbon materials and the effect on the catalytic properties.
3. Nitrogen is briefly introduced in the manuscript as a powerful heteroatom to increase the catalytic performance of carbon materials. However, it is widely known that not all N-doped carbon materials provide excellent ORR performance. Edge-type quaternary and pyridinic N are the two species that have proven to be active towards the ORR. This needs to be pointed out and discussed in the manuscript. I recommend supporting this statement with the following literature in DFT and experimental chemistry: Journal of the American Chemical Society 136(39), pp. 13629-13640; Journal of Physical Chemistry C, 2008, 112(38), pp. 14706–14709; Journal of Materials Chemistry A, 2019, 7(42), pp. 24239–24250;
4. While N is mentioned in the manuscript, there are other important heteroatoms that are not handled in the manuscript, despite entitling “heteroatom-doped” the manuscript. Please, more information about P, S, B and O functional groups is imperative. The authors can use the following literature to know the background in this topic: Carbon, 2020, 165, pp. 434–454.
5. In the synthesis method, the activation process is handled in the manuscript. However, for non-experts on this topic, it would be interesting to add a brief discussion about the differences between chemical and physical activation.
6. In the section “Application of heteroatom-doped porous carbon in Pt catalysts”, the authors need to highlight the Pt content in the recent Pt-based heteroatom-containing electrocatalysts to point out the novelty of these works. Otherwise, the reduction of the Pt content cannot be observed.
7. In the section “Summary and outlook”, the authors mentioned a series of problems that need to be solved. In the first point, the authors discussed that a problem to address is the stability of heteroatom-doped carbon materials in acidic conditions. However, the authors have not talked about it before in the manuscript. The authors need to explain this issue in the heteroatom-doped metal-free section. Moreover, they need to explain what is already written about it: Here is some important information about this issue that the authors need to talk about if they want to propose this statement as a problem to overcome: Carbon 2019, 148, 224−230.; ACS Applied Materials and Interfaces, 2020, 12(49), pp. 54815–54823; ACS Catal. 2015, 5, 4325−4332. ; Carbon, 2022, 189, pp. 548–560
Author Response

(The authors gave the same response as above.)

Round 2
Reviewer 1 Report
Authors have answered all the queries and the manuscript is improved.